# Hospitalization for Acute Respiratory Tract Infection in a Low-Antibiotic-Prescribing Setting: Cross-Sectional Data from General Practice

**DOI:** 10.3390/antibiotics9100653

**Published:** 2020-09-29

**Authors:** Christin Löffler, Attila Altiner, Annette Diener, Reinhard Berner, Gregor Feldmeier, Christian Helbig, Winfried V. Kern, Anna Köchling, Michaela Schmid, Gerhard Schön, Helmut Schröder, Karl Wegscheider, Anja Wollny

**Affiliations:** 1Institute of General Practice, Rostock University Medical Center, 18057 Rostock, Germany; altiner@med.uni-rostock.de (A.A.); annette.diener@web.de (A.D.); gregor.feldmeier@med.uni-rostock.de (G.F.); christian.helbig@med.uni-rostock.de (C.H.); anja.wollny@med.uni-rostock.de (A.W.); 2Department of Pediatrics, University Hospital Carl Gustav Carus, TU, 01307 Dresden, Germany; reinhard.berner@uniklinikum-dresden.de; 3Division of Infectious Diseases, Department of Medicine, Faculty of Medicine and Medical Center, University of Freiburg, 79085 Freiburg, Germany; winfried.kern@uniklinik-freiburg.de (W.V.K.); michaela.schmid@uniklinik-freiburg.de (M.S.); 4Clinic of Psychosomatic Medicine and Psychotherapy, Rostock University Medical Center, 18057 Rostock, Germany; anna.koechling@med.uni-rostock.de; 5Department of Medical Biometry and Epidemiology, University Medical Center Hamburg-Eppendorf, 20251 Hamburg, Germany; g.schoen@uke.de (G.S.); k.wegscheider@uke.de (K.W.); 6AOK Research Institute (WIdO), 10178 Berlin, Germany; helmut.schroeder@wido.bv.aok.de

**Keywords:** anti-bacterial agents, inappropriate prescribing, drug prescriptions, respiratory tract infections, primary health care

## Abstract

Background: Acute respiratory tract infections (ARTI) are the main cause of inappropriate antibiotic prescribing. To date, there is limited evidence concerning whether low levels of antibiotic prescribing may impact patient safety. We investigate whether antibiotic prescribing for patients seeking primary care for ARTI correlates with the odds for hospitalization. Methods: Analysis of patient baseline data (*n* = 3669) within a cluster-randomized controlled trial. Adult patients suffering from ARTI in German primary care are included. The main outcome measure is acute hospitalization for respiratory infection and for any acute disease from 0 to 42 days after initial consultation. Results: Neither the antibiotic status of individual patients (OR 0.91; 95% CI: 0.49 to 1.69; *p*-value = 0.769) nor the physician-specific antibiotic prescription rates for ARTI (OR 1.22; 95% CI: 1.00 to 1.49; *p*-value = 0.054) had a significant effect on hospitalization. The following factors increased the odds for hospitalization: patient’s age, the ARTI being defined as lower respiratory tract infections (such as bronchitis) by the physician, the physician’s perception of disease severity, and being cared for within group practices (versus treated in single-handed practices). Conclusions: In a low-antibiotic-prescribing primary care setting such as Germany, lack of treatment with antibiotics for ARTI did not result in higher odds for hospitalization in an adult population.

## 1. Introduction

In developed countries, acute respiratory tract infections (ARTI) are the leading reason for both consultations in primary care [1] and inappropriate antibiotic prescribing [2,3]. Across Europe, antibiotic prescriptions for ARTI range from 30% to 80%, with higher rates found in southern Europe and lower rates in central and northern Europe, including Germany [4,5]. Among the countries with a lower level of antibiotic consumption, Germany, however, stands out because of its comparatively high use of broad-spectrum antibiotic prescribing, indicating room for improvement [5]. Misuse and overuse of antibiotics is driving antibiotic resistance, with serious consequences for global health care, including fewer effective therapies for a growing number of infections, longer hospital stays, higher medical costs, and increased mortality [6,7]. Additionally, inappropriate antibiotic prescribing is associated with the occurrence of (avoidable) adverse drug reactions [8,9]. Concepts to explain the inappropriate use of antibiotics in ARTIs include physicians’ perceived safety when prescribing antibiotics, perceived patient pressure and conflict avoidance, and—in some rarer cases—inadequate knowledge among physicians [10,11,12]. 

Since the 1990s, several interventions to reduce inappropriate antibiotic prescribing for ARTI have been developed and evaluated. Approaches using shared decision making, communication skills training, and point-of-care testing have been most effective at reducing antibiotic prescribing for ARTI [13,14]. 

Today, there is limited evidence concerning whether very low levels of antibiotic prescribing may eventually impact on patient safety. Existing findings are contradictory: a British cohort study showed that low antibiotic prescribing correlated with a slight increase in the incidence of pneumonia and peritonsillar abscess [15]. Additionally, a retrospective study of aggregate data suggests an association between reductions in antibiotic prescribing for lower respiratory tract infections (LRTI) in general practice and an increase in pneumonia mortality in England and Wales [16]. The Genomics to Combat Resistance against Antibiotics in Community-acquired LRTI in Europe (GRACE) trial, however, showed no clear benefits of antibiotics [3]. Additionally, a recent British prospective cohort study assessing the effect of different antibiotic prescribing strategies for LRTI came to the conclusion that the immediate prescription of antibiotics may not reduce subsequent hospital admission or mortality for young people and adults with uncomplicated LRTI—though both are rare [17]. All these studies focus on the UK, where levels of antibiotic prescribing for ARTI reach almost 60% and are comparatively high. Much less is known about patient safety among ARTI patients in lower prescribing settings.

This paper therefore focuses on factors predicting hospitalization of adult patients in Germany presenting with ARTI in primary care, paying attention to the impact of antibiotic prescribing. In German primary care, there are no specific regulations impacting the prescription of antibiotics. Physicians are encouraged to apply relevant clinical guidelines when treating their patients. Mechanisms sanctioning high antibiotic prescribing do not exist. In this paper, we address the following questions: (a) does the individual prescription of antibiotics for ARTI correlate with the patient’s risk for hospitalization? And, (b) does the physician’s antibiotic prescribing rate for ARTI correlate with patients’ risk of hospitalization? 

## 2. Materials and Methods

### 2.1. Trial Design

We analyzed baseline data of the three-arm cluster randomized controlled CHANGE-2 trial. The trial measured antibiotic prescribing rates among primary care physicians for their patients presenting with symptoms of ARTI. The trial aimed at reducing unnecessary antibiotic prescribing by employing communication training and point-of-care testing (POCT) [18]. In CHANGE-2 patients from the regions Mecklenburg-Western Pomerania, Thuringia, Saxony, Brandenburg, Baden-Württemberg, and the city of Berlin were enrolled over three successive years. Baseline data were collected between January 2013 and May 2014. Cluster-randomization of participating practices and allocation to any intervention was performed only after that period. Thus, we used data of all three study groups before they were aware of group affiliation. Only at a later stage, participating practices were randomized by clusters, where each practice—be it a single-handed or a group practice—constituted a cluster. In summary, 105 primary care physicians were included at baseline. Participating primary care pediatricians (*n* = 87) were not included in this analysis. Criteria for patient inclusion in CHANGE-2 were as follows: Being insured with the German statutory health insurance company AOK, the provider of data. Within a typical primary care setting of the participating regional areas, at least 40% of all patients are insured with AOK. Despite some socio-economic differences, research showed that AOK data is adequate for analyzing patterns of health care utilization [19].A minimum age of 3 months, though children and teenagers up to 17 years of age were excluded from this analysis.A physician consultation visit due to an episode of ARTI according to the ICD classes J00-J04, J06, J13, J20, J22, whilst being otherwise healthy.

This definition included all typical ARTI such as bronchitis, tonsillopharyngitis (e.g., sore throat), and otitis media. Participants were required to give informed consent that included the acceptance of scientific use of relevant data stored by AOK. Patients with chronic pulmonary diseases, including asthma and COPD, and immune compromising diseases, including HIV and active neoplasms, were excluded from the study. Patients contributed to baseline data only once. In total, at baseline 3916 patients were included. 

In the follow-up of patients who consulted their primary care physician with symptoms of ARTI, trial data (patient and physician reported data) and health insurance data (routine care data) from primary and secondary care were merged. The major technical and legal challenges of the CHANGE-2 trial data merge are described elsewhere [18]. Thanks to this approach, there was no loss of patient data between enrolment and follow-up. Trial data included patient information on symptoms and duration of disease until consultation, as well as physicians’ assessment of the estimated severity of disease, and a working diagnosis. Information on symptoms and working diagnosis were gathered using free text. Data were categorized and coded after complete data collection. In contrast to disease codes assigned at recovery (e.g., at discharge from hospital), the working diagnosis was documented at the first contact with patients.

Health insurance data included physician data (specialization, region, level of urbanization, practice type, and inaugural year), patient demographic information (year of birth, sex, and date of death if applicable), relevant drug prescriptions (ATC codes), relevant medical outcomes (ICD codes), hospitalizations and corresponding diagnostic data, referrals, and information on sick leave. Data were retrieved for the period of one year before consultation until 42 days after enrolment into the study. 

The protocol was approved by the ethics committee of the Rostock University Medical Center before recruitment of physicians and patients on 10 September 2012, with the reference A 2012–0108. 

### 2.2. Data Analysis

Data analysis was based on consultations for ARTI for patients aged 18 and above. Trial data contained missing information for the covariates *severity of disease* (*n* = 48) and *duration of disease until consultation* (*n* = 205). Health insurance data did not contain any missing data. After exclusion of patients with missing data (*n* = 247), a total of 3669 patients from 105 physicians were analyzed. 

The outcome variable was defined as “beginning of hospitalization” from 0 to 42 days after consultation for ARTI. Hospitalizations for elective procedures were excluded from data analyses, while acute hospitalizations were included. In a second step, we analyzed acute hospitalizations labeled for respiratory infections only.

In total, eleven covariates entered multivariate modeling. These covariates were supposed to be relevant confounders and were set a priori. First, several patient-level covariates were included: physicians’ indicated variables, such as the working diagnoses *bronchitis*, *pneumonia*, and *viral infection*, each entering the model as dichotomous variable; *severity of disease* as dichotomous variable (mild or moderate mild vs. moderate severe or severe). Second, a number of patients’ indicated variables: *breathlessness* as dichotomous variable; and *duration of disease until consultation* in days as continuous variable. Next, *patient age* as continuous variable. Practice related covariates entered the model, too. These included *type of practice* (single-handed vs. group practice) and *physician specialization* (general practitioner, primary care internist). Last but not least, *physician’s specific antibiotic prescription rate for ARTI* as continuous variable, and *antibiotic prescription within initial consultation* as dichotomous variable were integrated. Physician’s specific antibiotic prescription rate for ARTI was calculated separately for each physician as percentage of their antibiotic prescriptions over all respective index consultations of participating patients. We performed a multilevel logistic regression model with physicians as random effect to analyze whether a consultation for ARTI lead to hospitalization or not. We reported odds ratios, confidence intervals for odds ratios, and the corresponding *p*-values. All analyses were computed by the statistical package R version 3.4.4. [20].

## 3. Results

Baseline data are presented in Table 1. Physicians’ specific antibiotic prescription rate for ARTI ranged from 0.0 to 64.4 prescriptions for 100 patients, with a median of 14.7 (see Figure 1). Median patient age was 40.8 years. This corresponds to the expected median age of patients visiting their GP for ARTI in this setting. The median age of patients hospitalized was 56.1 years; 43.7% of patients were male and 56.3% female. Median duration of disease until consultation was 5.1 days.

In a first step, we estimated the full multivariate regression model for the odds of hospitalization among adults presenting with ARTI in primary care integrating all covariates (*n* = 71 acute hospitalizations, see Table 2 for hospitalizations by condition and Table 3 for the model). Among these 71 hospitalizations, 55 patients had no prescription of antibiotics at the initial consultation, whereas 16 patients received antibiotics immediately. In this model, patients presenting with physician’s suspected diagnosis *bronchitis* had 75% higher odds to be admitted to hospital than patients without this diagnosis (95% CI: 1.03; 2.99; *p*-value = 0.039). Patients presenting with physician’s suspected *pneumonia* had no statistically significant higher odds for hospitalization than patients without such diagnosis (OR 1.51; 95% CI: 0.31; 7.33; *p*-value = 0.606). This finding might be attributed the low incidence of pneumonia in the sample. Physicians’ definition of disease as *moderate severe or severe* was associated with increased odds of hospitalization (OR 1.77; 95% CI: 1.04; 3.00; *p*-value = 0.035). The same was true for *patient age*: every 10-year increase in age resulted in 44% higher odds of hospitalization with ARTI (95% CI: 1.26; 1.65; *p*-value < 0.001). Compared to patients presenting with ARTI in single-handed practices, patients consulting physicians of group practices had 66% lower odds of hospitalization (OR 0.34; 95% CI: 0.13; 0.91; *p*-value = 0.031). According to the model, there was no significant effect of receiving an *antibiotic prescription within the initial consultation* (OR 0.91; 95% CI: 0.49; 1.69; *p*-value = 0.769). The impact of the *physician’s specific antibiotic prescription rate for ARTI* on hospitalization was close to significance: every 10% increase in the rate led to 22% higher odds of being admitted to hospital (OR 1.22; 95% CI: 1.00; 1.49; *p*-value = 0.054). Figure 2 summarizes the impact of covariates in the multivariate regression model on hospitalization.

In a further step, hospitalizations for reasons clearly related to ARTI (*n* = 18) were analyzed in the same way. From these 18 hospitalizations, 16 patients received no antibiotics at their initial consultation for ARTI, whereas 2 patients received antibiotics immediately. Additionally, 10 of these 18 patients were suffering from chronic comorbidities that supplemented hospitalization. Given the low number of events, compared to the first model, the second model is less robust. Nonetheless, the results are along the same lines: in the multivariate regression model integrating all covariates, patient age remains the only covariate showing significant estimates (OR 1.42; 95% CI: 1.03; 1.98; *p*-value = 0.034). After backward selection, patients with ARTI showing a *moderate severe or severe course of disease* face higher odds of hospitalization (OR 3.28; 95% CI: 1.19; 9.07; *p*-value = 0.022). Tables are not shown here. 

## 4. Discussion

Focusing on a low-antibiotic-prescribing primary care setting, this study showed that, overall, hospitalizations for ARTI were rare in our sample of more than 3600 patients suffering from ARTI. Whether antibiotics were prescribed during the initial consultation for ARTI had neither a significant nor a clinically relevant impact on hospitalization (OR 0.91; 95% CI: 0.49; 1.69; *p*-value = 0.769). Based on these data, a physician’s reluctance towards prescription of antibiotics for ARTI does not impact patient safety in our low-antibiotic-prescribing setting. 

We instead found that patients with a moderate severe or severe course of the disease (as defined by a physician) and older patients faced higher odds of being admitted to hospital when treated for ARTI. The same was true for patients with suspected bronchitis. It seems probable and is in line with expected clinical reasoning that, especially, patients with lower respiratory tract infections faced higher odds of being hospitalized. In case of bronchitis, our models clearly supported this assumption. The estimates for pneumonia pointed to the same direction without being significant, however. The low number of pneumonia cases might explain this finding. Being treated in group practices was associated with decreased odds of hospitalization. Possibly, these physicians may draw on their colleagues’ expertise and may spread responsibility across many shoulders. Evidence from another German study supports this view: physicians in group practices appreciate professional exchange and close teamwork [21].

Despite being one of the most common illnesses in primary care, so far there is only limited evidence on the benefits and risks of antibiotics for ARTI. The updated 2017 Cochrane review on antibiotics for bronchitis found no clear benefit of antibiotics for acute respiratory tract infections [22]. Additionally, with reference to risks, another Cochrane review—also published in 2017—investigated the effect of delayed prescribing on complications such as pneumonia, fever, or malaise. Though hospitalizations were not explicitly analyzed, the authors conclude that there is no evidence of differences in complication rates between immediate and delayed antibiotics or between delayed and no antibiotics [23]. Actually, so far only a few studies have investigated the impact of (reductions of) antibiotic prescribing on adverse events such as hospitalization. The few existing studies relied on time trend analyses of aggregated data [15,16]. However, matching individual patient data is not possible in this way. One exception is the study of Little et al. that focused on antibiotic prescribing for ARTI in the comparatively high antibiotic prescribing setting of the UK, where more than 60% of patients received an antibiotic for ARTI (compared to 15% in our study) [17]. To our knowledge, studies investigating risks of reduced antibiotic prescribing in low-antibiotic-prescribing settings do not exist.

One major strength of our study is the focus on an actual population in primary care. Given the design of the study, the data set had no dropouts. Another advantage of the trial is the matching of patient’s and physician’s reported trial data at the initial consultation, with real prescription data and hospital treatment data stemming from health insurance data. To minimize bias, data on symptoms as well as physician’s reported initial working diagnosis were collected at the time of the consultation and added to the data set. 

As to limitations, in the final data set the absolute number of hospitalizations was comparatively low with occasionally wide confidence intervals as a result. Given the rarity of hospitalizations for ARTI, future studies should include a higher number of patients. Second, though we took corrective action (such as the instruction to recruit all eligible patients consecutively), the recruitment of patients in participating practices might have induced selection bias. Additionally, we do not have data on patient compliance. Last but not least, data on symptoms and physician’s reported working diagnoses reflect individual clinical reasoning and are not objectifiable.

## 5. Conclusions

The most important implication derived from this analysis is that in this population, prescription of antibiotics during a consultation for ARTI had no significant effect on later hospitalization within a reasonable margin. Rather, patients’ age and clinical reasoning (perceived severity of disease and presence of a lower respiratory tract infection) were important predictors for hospitalization.

## Figures and Tables

**Figure 1 antibiotics-09-00653-f001:**
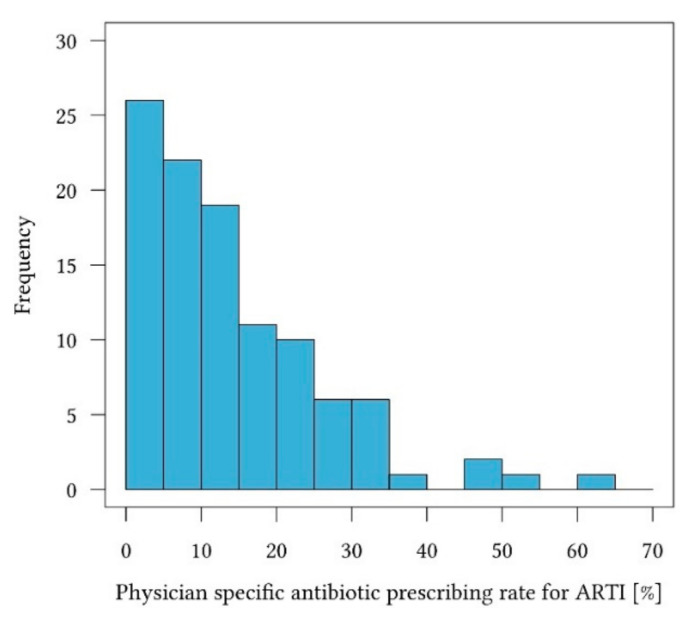
Frequency of physician specific antibiotic prescribing rate for acute respiratory tract infections, in terms of percentage.

**Figure 2 antibiotics-09-00653-f002:**
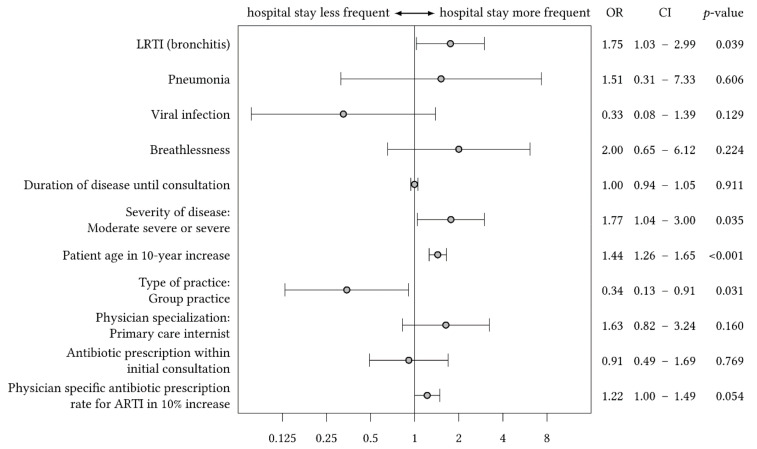
Forest plot of the full-multivariate regression model for the odds of hospitalization among patients consulting their physician because of ARTI, odds ratios, 95% confidence intervals, and *p*-values.

**Table 1 antibiotics-09-00653-t001:** Baseline data of 3669 patients included in the analyses, numbers, and percentages.

Variables	*n*	%
LRTI (bronchitis)	711	19.4
Pneumonia	34	0.9
Viral infection	436	11.9
Breathlessness	65	1.8
Severity of disease		
Mild or moderate mild	2466	67.2
Moderate severe or severe	1203	32.8
Type of practice		
Single-handed	2938	80.1
Group practice	731	19.9
Physician specialization		
General practitioner	3228	88.0
Primary care internist	441	12.0
Antibiotic prescription within initial consultation	569	15.5

**Table 2 antibiotics-09-00653-t002:** Hospitalizations by condition (*n* = 71).

Condition	Number
ARTI directly related (e.g., pneumonia)	18
Antibiotic prescribing related (e.g., enterocolitis due to Clostridium difficile)	1
Sepsis	1
Gastrointestinal diseases	4
Cardiovascular diseases	10
Musculoskeletal disorders and fractures	7
Other	30
**Total**	**71**

**Table 3 antibiotics-09-00653-t003:** Multivariate regression model for the odds of hospitalization among patients consulting. their physician because of ARTI (*n* = 71 hospitalizations), odds ratios, 95% confidence intervals, and *p*-values.

Variable	OR	2.5% CI	97.5% CI	*p*-Value
LRTI (bronchitis)	1.75	1.03	2.99	0.039
Pneumonia	1.51	0.31	7.33	0.606
Viral infection	0.33	0.08	1.39	0.129
Breathlessness	2.00	0.66	6.12	0.224
Duration of disease until consultation	1.00	0.94	1.05	0.911
Severity of disease				
Mild or moderate mild	1			
Moderate severe or severe	1.77	1.04	3.00	0.035
Patient age in 10-year increase	1.44	1.26	1.65	<0.001
Type of practice				
Single-handed	1			
Group practice	0.34	0.13	0.91	0.031
Physician specialization				
General practitioner	1			
Primary care internist	1.63	0.83	3.24	0.160
Antibiotic prescription within initial consultation	0.91	0.49	1.69	0.769
Physician specific antibiotic prescription rate for ARTI in 10% increase	1.22	1.00	1.49	0.054

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
