# Peer review of "Hospitalization for Acute Respiratory Tract Infection in a Low-Antibiotic-Prescribing Setting: Cross-Sectional Data from General Practice"

_antibiotics, 2020, doi:10.3390/antibiotics9100653_

Round 1
Reviewer 1 Report
The paper is well written with important information for readers in accordance with antibiotic stewardship guidelines.
Please mention clearly in the abstract and the text the period to discern hospitalization of patients after consultation.
Author Response
Reviewer 1
Please mention clearly in the abstract and the text the period to discern hospitalization of patients after consultation.
This information was present in the article in section 2.2, line 124: Data were retrieved for the period of one year before consultation until 42 days after enrolment into the study.
We added this information in the abstract, line 28: Main outcome measure is acute hospitalization for respiratory infection and for any acute disease from 0 to 42 days after initial consultation.
Reviewer 2 Report
Loffler et al. study the impact of antibiotic prescription on hospitalization in patients with ARTI. Author reports that antibiotic prescription status is not linked to hospitalization. Study is well executed but it has limitation given it is in Germany’s clinical system only. Author should address following:
Minor points:
- Is there any other such example studies in different clinical system (outside German clinical system) that supports this argument?
- Author should also comment on how well regulated is the antibiotic prescription in the clinical settings that are studied here?
- Please provide full description for short acronyms e.g. what is a GRACE trial?
Author Response
Reviewer 2
Is there any other such example studies in different clinical system (outside German clinical system) that supports this argument?
Unfortunately, there is very limited evidence on the effect of reductions of antibiotic prescribing for ARTI on complications such as hospitalizations. As stated in the article, existing evidence comes from the UK. To address this point, we added the following passage in the discussion, line 251: To our knowledge, studies investigating risks of reduced antibiotic prescribing in low antibiotic prescribing settings do not exist.
Also, we cite a recent Cochrane review investigating interventions of delayed antibiotic prescribing for ARTI and possible complications, line 242: Also, with reference to risks, another Cochrane review – also published in 2017 – investigated the effect of delayed prescribing on complications such as pneumonia, fever or malaise. Though hospitalizations were not explicitly analyzed, the authors conclude that there is no evidence of differences in complication rates between immediate and delayed antibiotics or between delayed and no antibiotics [23].
Again, trials included in the review focus mainly on the UK and USA.
Author should also comment on how well regulated is the antibiotic prescription in the clinical settings that are studied here?
To address this issue, we added the following information, line 74: In German primary care there are no specific regulations impacting the prescription of antibiotics. Physicians are encouraged to apply relevant clinical guidelines when treating their patients. Mechanisms sanctioning high antibiotic prescribing do not exist.
Please provide full description for short acronyms e.g. what is a GRACE trial?
This information has been added, line 64: The GRACE trial (Genomics to Combat Resistance against Antibiotics in Community-acquired LRTI in Europe), …
Reviewer 3 Report
Inappropriate antibiotic prescribing is a leading cause of the evolution of antibiotic-resistant bacterial strains. The present manuscript reports the result from a randomized controlled trial on adult patients at primary care setup in Germany. The focus of the study is to search for factors that predict the risk of hospitalization in patients introduced with ARTI and their relation to antibiotic prescription status. The important finding from the study is that hospitalization in patients brought with ARTI had no direct relation with the prescription of antibiotics but rather on the patients’ age and clinical reasoning. The paper has overall merit and importance to get published in Antibiotics. My comments are as follows,
- The main limitation of this study, the authors have mentioned in the manuscript though, is the low number of hospitalizations directly due to ARTI. Hence, in this scenario, it is especially difficult to draw a conclusive relationship between the rate of antibiotic prescription and its effect on hospitalization.
- Out of 18 hospitalization directly due to ARTI, how many were with the antibiotic prescription?
- In the hospitalized patients due to ARTI, how many of them had comorbidities? If yes, do these comorbidities supplement hospitalization?
- The title of the paper should mention 'low antibiotic prescription' or related terminology because the whole results and conclusions revolve around it.
Minor comments
- The title of Table 3 is incomplete.
Author Response
Reviewer 3
The main limitation of this study, the authors have mentioned in the manuscript though, is the low number of hospitalizations directly due to ARTI. Hence, in this scenario, it is especially difficult to draw a conclusive relationship between the rate of antibiotic prescription and its effect on hospitalization.
Yes, you are right and as you said, we mention this aspect in the discussion.
Out of 18 hospitalization directly due to ARTI, how many were with the antibiotic prescription?
We added this information in line 196: From these 18 hospitalizations, 16 patients received no antibiotics at their initial consultation for ARTI, whereas 2 patients received antibiotics immediately.
We also provide this information for n=71 hospitalizations, line 176: Among these 71 hospitalizations, 55 patients had no prescription of antibiotics at the initial consultation, whereas 16 patients received antibiotics immediately.
In the hospitalized patients due to ARTI, how many of them had comorbidities? If yes, do these comorbidities supplement hospitalization?
This information can now be found in line 197: Also, 10 of these 18 patients were suffering from chronic comorbidities that supplemented hospitalization.
The title of the paper should mention 'low antibiotic prescription' or related terminology because the whole results and conclusions revolve around it.
We changed the title into: Hospitalization for Acute Respiratory Tract Infection in a Low Antibiotic Prescribing Setting: Cross-sectional Data from General Practice
The title of Table 3 is incomplete.
Sorry, but in our version the title of table 3 is complete. Can you please specify? The complete title is, line 210:
Table 3. Multivariate regression model for the odds of hospitalization among patients consulting their physician because of ARTI (n=71 hospitalizations), odds ratios, 95% confidence intervals and p-values.